# Meeting 24-Hour Movement and Dietary Guidelines: Prevalence, Correlates and Association with Weight Status among Children and Adolescents: A National Cross-Sectional Study in China

**DOI:** 10.3390/nu14142822

**Published:** 2022-07-08

**Authors:** Yide Yang, Shuqian Yuan, Qiao Liu, Feifei Li, Yanhui Dong, Bin Dong, Zhiyong Zou, Jun Ma, Julien S. Baker, Xianxiong Li, Wei Liang

**Affiliations:** 1Department of Child and Adolescent Health, School of Medicine, Hunan Normal University, Changsha 410006, China; yangyide@hunnu.edu.cn (Y.Y.); shuqiany@hunnu.edu.cn (S.Y.); liuqiao1234567899@163.com (Q.L.); 2Key Laboratory of Molecular Epidemiology of Hunan Province, School of Medicine, Hunan Normal University, Changsha 410006, China; 3Institute of Child and Adolescent Health, School of Public Health, Peking University Health Science Center, Beijing 100191, China; bindong@bjmu.edu.cn (B.D.); harveyzou2002@bjmu.edu.cn (Z.Z.); majunt@bjmu.edu.cn (J.M.); 4Centre for Health and Exercise Science Research, Hong Kong Baptist University, Hong Kong, China; lifeifei@hkbu.edu.hk (F.L.); jsbaker@hkbu.edu.hk (J.S.B.); 5Department of Sport, Physical Education and Health, Hong Kong Baptist University, Hong Kong, China; 6School of Physical Education, Hunan Normal University, Changsha 410081, China

**Keywords:** obesity, overweight, underweight, physical activity, screen time, sleep duration, diet, 24 h movement

## Abstract

China is confronted with a “double burden” of underweight and overweight/obesity in children and adolescents. This study aimed to investigate the prevalence and correlates of meeting 24 h movement and dietary guidelines among Chinese children and adolescents. Further, the study aimed to examine the association of meeting 24 h movement and dietary guidelines with weight status in Chinese children and adolescents. A total of 34,887 Chinese children and adolescents were involved. Only 2.1% of participants met the 24 h movement guidelines. Compared to those who met all three 24 h movement guidelines, those who only met the sleep duration guideline was significantly associated with a higher risk of underweight (*p* < 0.05), and those who only met the moderate-to-vigorous physical activity, or screen time guidelines were significantly associated with a higher risk of overweight/obesity (*p* < 0.05). Compared with those meeting the dietary guidelines, those who did not meet the soft drink intake guideline had a significantly lower risk of underweight (*p* < 0.05), those who did not meet the fruit intake guideline had a significantly lower risk of overweight/obesity (*p* < 0.05), and those who did not meet the milk intake guideline showed a significantly higher risk of overweight/obesity (*p* < 0.001). These findings indicate a significant association between meeting the 24 h movement and dietary guidelines and weight status among Chinese children and adolescents.

## 1. Introduction

Overweight and obesity in children and adolescents have been a worldwide public health problem [1]. Especially in China, childhood and adolescent obesity has been increasing at an alarming rate. A recent health report from The State Council Information Office of the People’s Republic of China (2020 Dec) has shown that the prevalence of obesity among Chinese children aged 6–17 years was about 20% during 2014–2019 (http://www.scio.gov.cn/xwfbh/xwbfbh/wqfbh/42311/44583/wz44585/Document/1695276/1695276.htm; accessed on 7 July 2022). Childhood and adolescent obesity have been proven to be significantly associated with metabolic and cardiovascular diseases in adulthood [2]. Previous studies have also shown that a higher body mass index (BMI) during adolescence was associated with a higher risk of malignancies in adulthood (e.g., breast cancer, colorectal cancer) [3,4]. In addition, underweight of children and adolescents is also a worldwide public health challenge, especially in developing countries, such as China [5]. Previous studies have shown a significant association between underweight and higher risk of infectious diseases among children and adolescents [6]. Underweight, overweight and obesity among children and adolescents are all detrimental to health throughout the whole life course [7]. As one of the low-income and middle-income countries, China is facing a complex shift from the predominance of undernutrition to dual/double burden of overnutrition and undernutrition [8,9].

Weight status of children is closely associated with healthy lifestyle behaviors, such as physical activity (PA), sedentary behavior in front of a screen, sleep and dietary behaviors [10,11,12]. Regular PA and adequate sleep (SLP) have a protective effect on healthy weight status among children and adolescents, while extended screen time (ST) generally has a negative impact [10,11,12]. Previous studies indicated that altering physical activity and sedentary behavior was effective in improving overweight and obesity in children and adolescents [13,14]. Additionally, a previous study also found a significantly lower level of physical activity in underweight boys compared with normal weight [15]. Alongside the individual benefits, the combined effects of these movement behaviors in a 24 h day on health have stimulated increasing concerns recently [10,11,12,16]. The Canadian 24-Hour Movement Guidelines integrated recommendations that respect the intuitive and integrative nature of children’s physical activity [17]. The Guidelines recommend that school-aged children (5–17 years) should perform at least 60 min of moderate-to-vigorous physical activity (MVPA) per day, sleep 9 to 11 h per day and keep screen behavior to less than 2 h per day [17]. The process of developing the guidelines was guided by the Guidelines for Research Evaluation (AGREE) II instrument, and systematic reviews of evidence informing the guidelines were assessed according to the Grading of Recommendations Assessment, Development and Evaluation (GRADE) approach. Four systematic reviews (physical activity, sedentary behavior, sleep, integrated behaviors) and several health indicators were recognized by expert consensus [16]. The Canadian 24-Hour Movement Guidelines integrated recommendations have been applied as the research paradigm [18] and have been used by many researchers in their research [10,19]. Previous studies have reported that children and adolescents who met the 24-hour movement guidelines had a lower risk of metabolic and cardiovascular diseases than those who did not [10,20]. Meeting 24-hour movement guidelines has proven to be associated with a lower z-score for BMI [21]. Previous studies have also shown a low prevalence of meeting the 24 h movement guidelines among children and adolescents globally. In Hong Kong, China, the prevalence of meeting the 24 h movement guidelines was only 1% [22]. Similar results were reported in Canada, USA and the mainland of China [21,23].

Dietary behaviors have also been demonstrated to be important determinants of weight status [24]. Eating too many high-calorie foods (such as fast food and soft drink) is associated with an increased risk of obesity [25]. Children who have an unbalanced diet or picky eating behavior are more likely to be underweight or malnourished [26]. The World Health Organization (WHO) issued relevant dietary guidelines and patterns, recommending a reduction in free sugars to less than 5% of total energy intake, eating at least five portions or 400 g of fruits and vegetables per day and having breakfast and drinking milk every day for additional health benefits [27]. Analysis of dietary patterns has been widely used to explore the relationships between diet and chronic diseases, helping to develop appropriate weight recommendations [28,29]. Previous studies indicated a poor meeting of WHO diet recommendations among children and adolescents and showed that dietary patterns were closely associated with the prevalence of childhood obesity, subsequently leading to numerous negative consequences [12,30]. 

A growing body of epidemiological and empirical evidence has emphasized the combined effect of movement behaviors and healthy diet on weight status and related health domains [11,31,32,33]. However, most previous studies focused only on the individual effects of these behaviors on weight status in children and adolescents [13,34]. In addition, previous studies have paid more attention to overweight and obesity in children and adolescents, while the underweight status has been comparatively ignored [26]. To the best of our knowledge, only one national survey investigated meeting the 24 h movement guidelines among Chinese children and adolescents (grade 4–12), whereas children at grade 1–3 (aged 6–10 years) were not involved, and the relationships between 24 h movement and underweight status were not examined [17]. Further, there is limited evidence on the meeting of dietary guidelines among Chinese children and adolescents, especially using a national-level survey. There is also a scarcity of national data examining the combined effects of 24 h movement and dietary behaviors on comprehensive weight status among children and adolescents. 

Therefore, the current study aimed to (1) investigate the prevalence of complying with 24 h movement and dietary guidelines among Chinese children and adolescents (6–17 years old); (2) identify the correlates of each behavior; and (3) examine the association between meeting 24 h movement and dietary guidelines and weight status (including underweight and overweight/obesity) among Chinese children and adolescents. 

## 2. Methods

### 2.1. Participants

The present study was embedded in the baseline survey of a nationwide health lifestyle intervention program (Clinical Trial Registration website and date: https://www.clinicaltrials.gov/, accessed on 22 January 2015; Registration No: NCT02343588). The details of the study design and key findings of this national program have been published elsewhere [35,36]. In brief, this is a national representative survey using a multi-stage random cluster sampling design. We recruited over 70,000 children and adolescents aged 6–18 years from 94 schools (including primary, secondary and high school) in seven provinces or municipalities (Hunan, Liaoning, Ningxia, Guangdong, Shanghai, Chongqing and Tianjin) in mainland China. More details of the selection process are outlined in Figure 1. From the response sample (*n* = 65,347), a total of 34,887 participants were included in the final analysis. This study followed the guidelines of the STROBE checklist and was approved by the Medical Ethical Committee of Peking University (No. IRB0000105213034). Signed informed consent forms were obtained from both students and their parents.

### 2.2. Measurement

#### 2.2.1. Demographic Information

Demographic information, including age, sex, residence (rural/urban), region (central China: Hunan; northeast China: Liaoning; northwest China: Ningxia; south China: Guangdong; east China: Shanghai; southwest China: Chongqing; north China: Tianjin), grade (primary school/secondary school/high school), ethnicity (Han/non-Han), was self-reported by students. Socioeconomic information, including family structure (single child or not), parents’ age (≤30 years, 30–40 years, 40–50 years, >50 years), parents’ education level (primary and below, secondary or high school, college or above), monthly household income (<CNY 5000, CNY 5000–12,000, ≥CNY 12,000), parents’ weight status (underweight, normal weight or overweight/obesity), was self-reported by students’ parents.

#### 2.2.2. 24 h Movement Behaviors

The 24 h movement behaviors were self-reported by the students and their parents together. PA was measured by reliable and valid items derived from a previous study in Chinese elementary students (reliability coefficient = 0.82) [37,38]. Frequency (days each week) and duration (specific hours spent on these activities on each of these days) of vigorous PA (obviously increasing one’s breathing and heart rate, such as running, basketball and football, etc.) and moderate PA (to some extent increasing one’s breathing and heart rate, such as table tennis, biking, badminton, etc.) were investigated [39]. Participants were required to record physical activity (PA) in and out of school for seven consecutive days. Participants who met or who did not meet ≥60 min moderate-to-vigorous physical activity (MVPA) per day for seven consecutive days were coded as 1 and 0, respectively [16].

Children and adolescents’ sleep duration (SLP) was assessed by one item from the China Health and Nutrition Survey, which has acceptable validation (reliability coefficient = 0.83). The item asked participants to report how long they slept each day for the previous week. The choices were “<7 h”, “7~9 h”, “9~11 h”, “>11 h”. Students aged 5 to 13 who slept for 9 to 11 h at night and those aged 14 to 17 who slept for 8 to 10 h at night, and maintained a consistent bedtime and wake time, met the SLP guidelines [16].

Screen time (ST) was assessed using reliable and valid items derived from a previous study in Chinese school-aged children (reliability coefficients = 0.72–0.74) [30], based on how much time they spent on watching TV, using computers or playing video games at school and home in the past seven days. Screen time ≤2 h is considered as meeting the ST guidelines [16].

#### 2.2.3. Dietary Behaviors

Dietary behaviors were measured using reliable and valid items derived from China Health and Nutrition Survey (CHNS) [40]. Eating behaviors mainly include breakfast, fruits, vegetables, meat, milk and soft drinks. The students and their parents were asked to answer the questions, e.g., “how often (days of the week) and how much (consumption) of fruits, vegetables and meat did the child/children consume in the past week”. Average daily intake of fruits, vegetables and meat was estimated as follows: Average daily intake = [frequency × (number of days per day)]/7 [41]. For the intake of soft drinks, each child was asked to answer two questions, namely the frequency of all soft drinks consumed (number of cups per week) and usual serving size (cup), such as Coca-Cola, Sprite, Nutrition Express, orange juice and Red Bull (one serving = 250 mL). We used these items to calculate children’s intake of soft drinks, breakfast and milk consumption during the past 7 days [39,42].

Eating fruit ≥2 serving/day, eating vegetables ≥4 serving/day, eating meat ≥2 serving/day, soft drink intake = 0, eating breakfast every day, drinking milk every day, were considered to be meeting the dietary guidelines. Eating fruit <2 serving/day, eating vegetables <4 serving/day, eating meat <2 serving/day, soft drink intake ≥1 time, not eating breakfast everyday = 0, milk intake = 0, were considered not to be meeting the dietary guidelines.

#### 2.2.4. Weight Status

Trained technicians used standardized protocols to measure the children’s height and weight. Each participant’s height and weight were objectively measured twice, and the average height and weight were used to calculate the BMI. BMI is calculated by dividing weight in kilograms by height in meters squared (kg/m^2^). Children’s weight status was classified as underweight, normal weight and overweight/obese based on sex- and age-specific BMI cutoff points provided by the International Obesity Task Force (IOTF) [43]. Parents’ BMI was calculated from the height and weight provided in the questionnaire. The weight status of parents was determined using a Chinese adult standard [44]. Overweight was defined as BMI ≥24 kg/m^2^ ~ <28 kg/m^2^ and BMI ≥28 kg/m^2^ was defined as obese.

### 2.3. Statistical Analysis

Descriptive statistics were reported using mean and standard deviations (SD) or percentage (%). *t*-tests were used for evaluating the differences between continuous variables, and the Chi-square test was used for differences between categorical variables. Frequency analyses were used to obtain the prevalence of not meeting the different 24 h movement guidelines, their combination (i.e., MVPA, ST, SLP, MVPA+ST, MVPA+SLP, ST+SLP, All) and prevalence of not meeting the dietary guidelines (i.e., breakfast, fruit, vegetable, meat intake, soft drink intake and milk intake). Logistic regression models were used to examine the associations between meeting the behavioral guidelines and weight status in children and adolescents. Statistical significance level was set at *p* < 0.05 (two-tailed). All data were analyzed using IBM SPSS for Windows (version 27.0, SPSS Inc. Chicago, IL, USA).

## 3. Results

### 3.1. Sample Characteristics

Descriptive characteristics of the study sample are presented in Table 1. The mean age of the participants was 11.42 ± 3.22 years, and 16,813 participants (48.2%) were girls. The mean BMI was 18.76 ± 3.84 kg/m^2^, with significant sex differences (*p* < 0.001). The proportions of the participants in primary, secondary and high school were 53.5%, 25.6% and 20.9%. Most of the sample (94.7%) were of Han ethnicity. A total of 45% of the participants were based in urban residence. Single children comprised 68.7% of the total sample, with a significant sex difference (*p* < 0.001). Only 26.9% and 24.5% of the participants’ fathers and mothers received education at college level or above. About 9.2% of the families reported a monthly household income of no less than CNY 12,000. About 9.6% of the participants reported a family history of cardiovascular diseases. The proportion of overweight or obese children was 22.1%, with a significant sex difference (27.1% in boys vs. 16.70% in girls, *p* < 0.001). For underweight, the proportion in the total sample was 11.3%, with a significant sex difference (9.4% in boys vs. 13.50% in girls, *p* < 0.001). For parents, about 41.1% (fathers) and 20.1% (mothers) were overweight or obese, while 1.7% (fathers) and 3.7% (mothers) were underweight. Figure 2 shows the prevalence for overweight/obesity, underweight and not meeting the 24 h movement guidelines among children and adolescents in seven provinces of China. Tianjin and Liaoning had the highest prevalence of obesity (>20%), Guangdong had the highest prevalence of underweight (>16%), Ningxia had the highest prevalence of not meeting the 24 h movement guidelines (>9%).

### 3.2. Prevalence and Correlates of Meeting 24 h Movement Guidelines

Table 2 shows the prevalence of children and adolescents meeting the different 24 h movement guidelines, their combinations and various dietary guideline components. The prevalence of meeting the 24 h movement guidelines was 2.1% (95%CI: 1.9%, 2.2%), with significant sex difference and age difference (all *p* < 0.001). The percentage of meeting ST, MVPA and SLP guidelines was 17.8%, 35.7% and 36.4%, respectively, with a significant sex difference (*p* < 0.001). There is also a significant age difference in the prevalence of meeting the ST guideline (*p* < 0.001). The prevalence of meeting two components of 24 h movement (MVPA+ST, MVPA+SLP, ST+SLP) was 5.6%, 13.3% and 6.3%, respectively, with a significant sex difference (*p* < 0.001). For the prevalence of meeting MVPA+ST, MVPA+SLP guidelines, there was also a significant age difference (*p* < 0.05).

The prevalence of meeting different components of dietary guidelines (including breakfast, fruit intake, vegetable intake, meat intake, soft drink and milk intake) varied prominently, with 84.4%, 26.1%, 8.2%, 21.1%, 32.2% and 44.1%, respectively (Table 3). For breakfast, fruit, soft drink and milk intake guideline, 6–11-year-old children reported significantly higher meeting prevalence than the 12–17-year-old adolescents (*p* < 0.001). However, for vegetable and meat intake, 12–17-year-old adolescents had a higher meeting prevalence than 6–11-year-old children (*p* < 0.001). In terms of the sex difference, boys reported significantly higher meeting prevalence of vegetable, meat and milk intake guidelines (*p* < 0.001), while girls had higher prevalence of meeting fruit and soft drink intake guidelines (*p* < 0.01).

Figure 3 and Figure 4 present the correlates of meeting the 24 h movement and dietary guidelines. Results revealed that grade, sex, residence, mother’s age and father’s education level are significant correlates of the child meeting the 24 h movement guidelines. For meeting dietary guidelines among children and adolescents, significant correlates included grade, residence, monthly household income, father’s age, mother’s age and mother’s education level.

### 3.3. Relationships between Meeting 24 h Movement and Dietary Guidelines and Weight Status

Table 4 demonstrates the relationships between meeting different 24 h movement guidelines/dietary guidelines and the risk of underweight and overweight/obesity among children and adolescents. With adjustment for age, gender, grade, parental education level, parents’ age, family income, family chronic disease history, residence, location, family structure and parents’ weight status and movement variables in model 1, meeting none of the guidelines, only meeting the MVPA or ST guideline had a significant association with higher risk of overweight/obesity than meeting all three (MVPA+SLP+ST) guidelines (None (OR = 1.26, 95%CI = 1.03,1.54, *p* < 0.05),MVPA (OR = 1.33, 95%CI = 1.09,1.64, *p* < 0.05), ST (OR = 1.35, 95%CI = 1.09,1.67, *p* < 0.05). Only meeting the SLP guideline showed a statistically significant association with significantly higher risk of underweight than meeting all three 24 h movement guidelines (OR = 1.35, 95%CI = 1.04,1.75, *p* < 0.05). After further adjustment of dietary variables in model 2, the results for underweight almost did not change. Results for underweight were also similar. For overweight/obesity, only meeting MVPA (OR = 1.30, 95%CI = 1.06, 1.60) and ST (OR = 1.32, 95%CI = 1.06,1.64) exhibited a statistically significant association than meeting all three 24 h movement guidelines (*p* < 0.05).

For dietary guideline components in model 2 (further adjustment of movement variables on the basis of model 1), participants who did not meet the soft drink guideline presented a significant association with decreased risk of underweight (OR = 0.91, 95%CI = 0.84, 0.98) compared to those meeting it. Those who did not meet the fruit intake guideline were significantly associated with lower risk than those meeting the fruit intake guideline (OR = 0.93, 95%CI = 0.87, 0.99, *p* < 0.05). Those who did not meet the milk intake guideline had a significantly higher risk of overweight or obesity than those meeting the milk guideline (OR=1.16, 95%CI=1.10, 1.23, *p* < 0.001).

## 4. Discussion

To the best of our knowledge, this is the first national cross-sectional study to provide comprehensive evidence for both 24 h movement and dietary behaviors, in terms of the prevalence and correlates of meeting the guidelines, and the association of guideline meeting with weight status (both overnutrition and malnutrition) among Chinese children and adolescents.

We found that, in China, Tianjin and Liaoning had the highest prevalence of overweight/obesity, Guangdong had the highest prevalence of underweight, Ningxia had the highest prevalence of not meeting the 24 h movement guidelines. Economic and dietary differences between the different regions in China may explain these findings. In China, the economic development level between urban and rural populations is different, and this difference also exists in China’s eastern and western regions. Children in urban or eastern areas have a higher socioeconomic status [45]. A previous study found that China’s economic development has a significant effect on public health, while this effect is heterogeneous at the regional level [46]. Western China (including Ningxia) is least developed, which may lead to a negative impact on the effectiveness of public health policies, such as 24 h movement and dietary guidelines. In addition, China national surveys have demonstrated a higher consumption of energy-dense foods in children with a high socioeconomic status [47]. Moreover, the characteristics of regional diets in China are also different significantly [48]. Southern dietary patterns are characterized by a high intake of rice, meat, vegetables and poultry. Residents of Guangdong province in southern China not only follow the traditional southern diet but also prefer food with less salt and oil, and cooking by steaming and boiling, which may lead to weight loss [49], whereas northern dietary patterns (including Ningxia and Tianjin and Liaoning) are characterized by a high intake of soybeans and wheat, which are more likely to lead to overweight/obesity [48]. The results of our study showed that Tianjin (north China) and Liaoning (northeast China) had the highest prevalence of overweight/obesity, while Guangdong (south China) had the highest prevalence of underweight, which is in line with China’s national conditions.

The Chinese government has formulated a rural vitalization policy and anti-poverty relocation and settlement program (ARSP) to narrow the economic development gap between urban and rural areas and solve the poverty trap in western regions [50,51], helping reduce the risk of underweight among children and teenagers in poor areas. In addition, the prevalence of obesity is rising among children and adolescents in developed regions [52]. Therefore, a panel of Chinese experts developed a consensus statement, which stressed the importance of promoting physical activity and provided suggestions for promoting physical activity among school-aged children and adolescents to prevent childhood and adolescent diseases, including obesity, and achieve the Healthy China 2030 goals [53]. The Healthy China 2030 national strategy promotes the most remarkable measures to control the obesity and underweight problems, and it was initiated in 2016 [5]. Over the past decades, China has made many efforts to tackle the obesity and underweight problems [54]. Moreover, the Dietary Guidelines for Chinese (DGC-2016) were released in 2016, with the aim of developing dietary guidelines suitable for Chinese people and helping them develop healthier and more scientific eating habits [55,56].

We found that the prevalence of meeting all three 24 h movement recommendations was only 2.1%. In addition, those meeting none, only one or two of the 24 h movement guidelines had a stronger association with underweight and also overweight/obesity than those meeting all movement guidelines. Those children who did not meet the soft drink guidelines presented a significant association with a lower risk of underweight than those meeting the recommendation. Meanwhile, those children who did not meet the milk intake guidelines were associated with an increased risk of underweight than those who met the milk intake guidelines. For overweight/obesity, those who did not meet the fruit intake guidelines reported a significant association with a lower risk than those meeting the fruit intake guidelines; those who did not meet the milk intake guidelines had a significantly stronger association with overweight or obesity than those who met the milk guidelines.

Our findings are consistent with most previous studies. Previous studies conducted in family-based settings indicated that altering dietary habits, PA and ST was effective in improving overweight and obesity in children and adolescents [14,57,58]. School-based PA interventions have also shown a salient effectiveness in preventing obesity in children and adolescents [13]. A longitudinal study conducted among Danish children found that PA, ST and SLP durations were significantly associated with cardio-metabolic risk in childhood [59]. In addition, as the results in this study show, combined associations were evident between ST/MVPA/SLP and the metabolic syndrome (MetS). These findings are in line with the results of our study, which found that those meeting none, only one or two of the movement guidelines were significantly associated with higher risk of overweight/obesity than those meeting all the 24 h movement guidelines. In China, previous studies have focused on the relationship between a single behavior and overweight and obesity in children [60,61]. A recent study analyzed the association of the combination of PA, video behavior and sleep standards with overweight and obesity in children and adolescents [62], showing that reasonable sleep duration can help reduce the risk of overweight and obesity. However, studies that investigated the association between combined 24 h movement and dietary behaviors and weight status (including underweight and overweight/obesity) in children and adolescents are rare in China, particularly using nationally representative samples. A previous study conducted in Poland found a significantly lower level of physical activity in underweight boys compared with normal weight. However, underweight was not associated with ST [15]. Additionally, a study conducted in Cambodia found a positive relationship between short sleep time and underweight among school students [63]. However, some studies indicated that short sleep time had inverse relationships with underweight [64]. Our study found that only meeting the SLP guidelines was significantly associated with underweight compared to meeting all the guidelines. A previous study conducted in Quebec children observed a U-shaped relationship between all adiposity indices and sleep duration. This study found that children who slept 11 to 11.9 h per night had lower total energy intake and fat intake per day than those who slept shorter or longer [65]. Additionally, children who slept 11 to 11.9 h per night had lower BMI scores and percentage of body fat [64]. This might provide an explanation for our research findings, hinting that specific sleep time ranges should be targeted to prevent both underweight and overweight/obesity.

Previous epidemiological evidence has indicated that consuming adequate fruit or vegetables is inversely associated with obesity; low fruit or vegetable intake may lead to overweight/obesity [66,67,68,69]. However, our study shows that compared with participants meeting the fruit intake guidelines, those who did not meet the fruit intake guidelines showed significantly lower risks of overweight/obesity. Most previous studies did not distinguish the ways of consuming fruits, such as fresh whole fruits, 100% fruit juice, blended juice or tinned fruit, leading to inconsistent results on the association between fruit consumption and weight [70,71,72,73]. Fruit contains various kinds of simple sugars (glucose, fructose, etc.), which are well known to lead to obesity [74,75,76]. In addition, studies have confirmed that tinned fruit and blended juice contain added sugars, which may increase the risk of obesity [77]. These findings support our results. Additionally, previous studies found that low socioeconomic status and fruit intakes were associated with adolescent overweight/obesity [73,78]. Fruit intakes mediated the relationship between low socioeconomic status and adolescent overweight/obesity [78]. These findings deserve further exploration in the future. The risk of underweight was found significantly decreased in children and adolescents who consumed soft drinks in the study. This finding is similar with the previous study, where those adolescents who consumed soft drinks had a 73% lower risk of underweight [79]. The possible mechanisms of this finding can be demonstrated by virtue of soft drinks’ high added sugar content, low satiety and low compensation for total energy [79,80,81]. In our study, those who did not meet the milk intake guideline had a significantly increased risk of overweight/obesity. Milk was classified as whole-fat, reduced-fat, low-fat and skim milk; whole milk represents a significant source of total and saturated fat [82]. Therefore, some studies indicated that whole milk intake may contribute to excess energy and fat intake, which leads to obesity [82,83]. However, a previous study examined the percentage of children and adolescents who consumed different types of milk and found that more than 60 percent of children and adolescents chose low-fat or skim milk, and most overweight individuals did not choose whole milk [84]. In addition, a latest double-blind randomized study showed that whole milk intake does not increase the risk of obesity [85]. Therefore, the effect of fat in milk on obesity is negligible. Moreover, some studies have demonstrated that low milk intake in childhood is a risk factor for obesity, which is similar to our results [86,87,88]. Several possible mechanisms may explain these results, including milk intake being associated with greater satiety and dairy being associated with increases in fat oxidation [89,90]. The importance of the calcium content of dairy in regulating body fat storage has been demonstrated [91]. Additionally, branched-chain amino acids leucine of dairy may play a significant role in the partitioning of dietary energy [92].

At present, limited studies have investigated the guideline meeting situation for both 24 h movement and dietary guidelines and their correlates and associations with weight status among children and adolescents, especially in China. Our findings may bear considerable implications for future research and health practice, underlining the importance of the combined effect of diverse lifestyle behaviors on children’s and adolescents’ weight status. Although the data used in this study were collected during 2014 and 2015, they could contribute to estimating the trend change of children’s and adolescents’ practice of comprehensive lifestyle behaviors by adding to the representative evidence for future intervention and policymaking. Despite the implications of this paper, several limitations should be noted. Firstly, even though we used nationally representative samples, the cross-sectional design may limit the explanation of causal relationships between study variables. Moreover, all the behavioral indicators were self-reported, and this may lead to recall bias.

## 5. Conclusions

The prevalence of complying with the 24 h movement and dietary guidelines among Chinese children and adolescents was low. Diverse sociodemographic variables were significant correlates of meeting the 24 h movement and dietary guidelines. Meeting the 24 h movement and dietary guidelines was associated with weight status in Chinese children and adolescents. The study findings provide empirical evidence for future health interventions and policymaking in the fight against childhood overnutrition and malnutrition.

## Figures and Tables

**Figure 1 nutrients-14-02822-f001:**
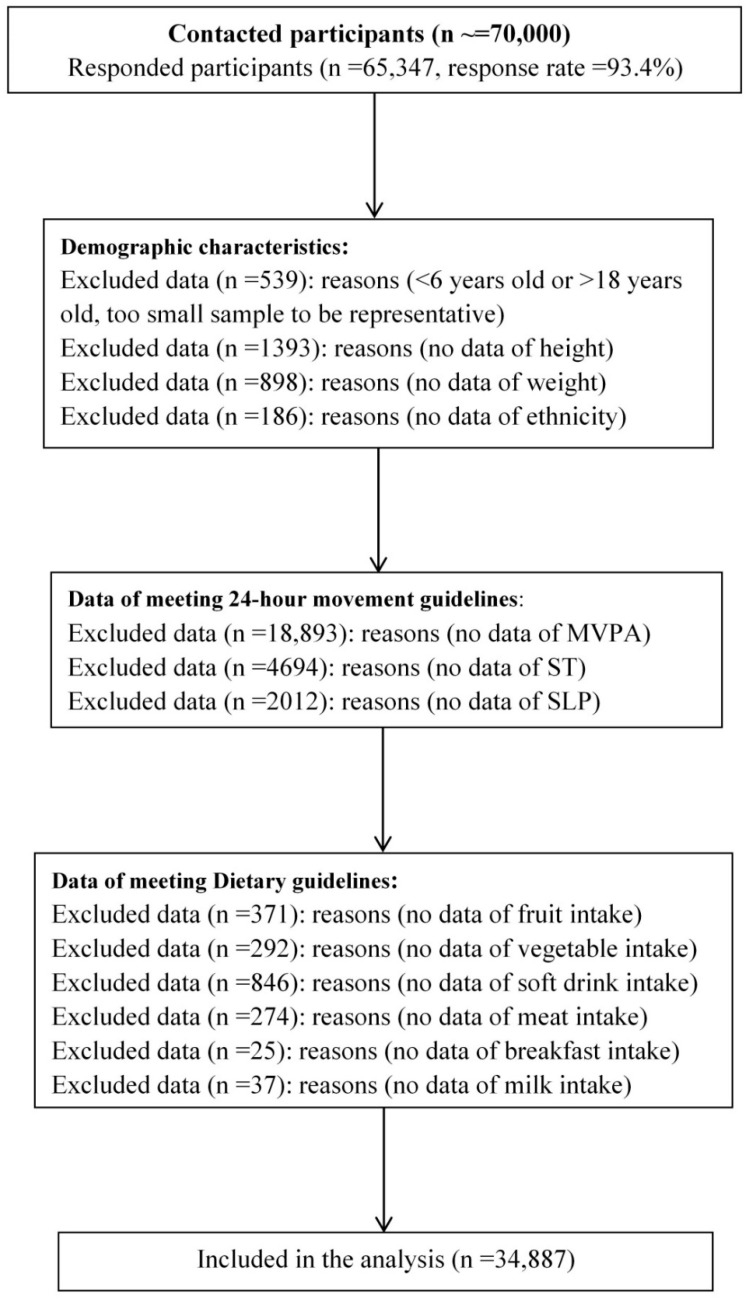
The procedure of obtaining final study sample in the present study. (MVPA = moderate-to-vigorous physical activity; ST = screen time; SLP = sleep duration).

**Figure 2 nutrients-14-02822-f002:**
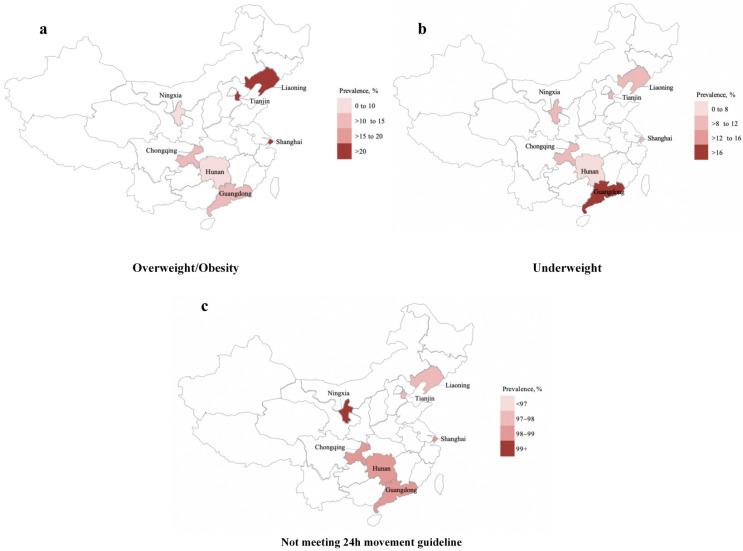
Prevalence of underweight, overweight or obesity and meeting the 24 h movement guidelines among school-aged children and adolescents in mainland of China (**a**) presents the prevalence of overweight/obesity, (**b**) presents underweight and (**c**) presents not meeting 24 h movement guidelines.

**Figure 3 nutrients-14-02822-f003:**
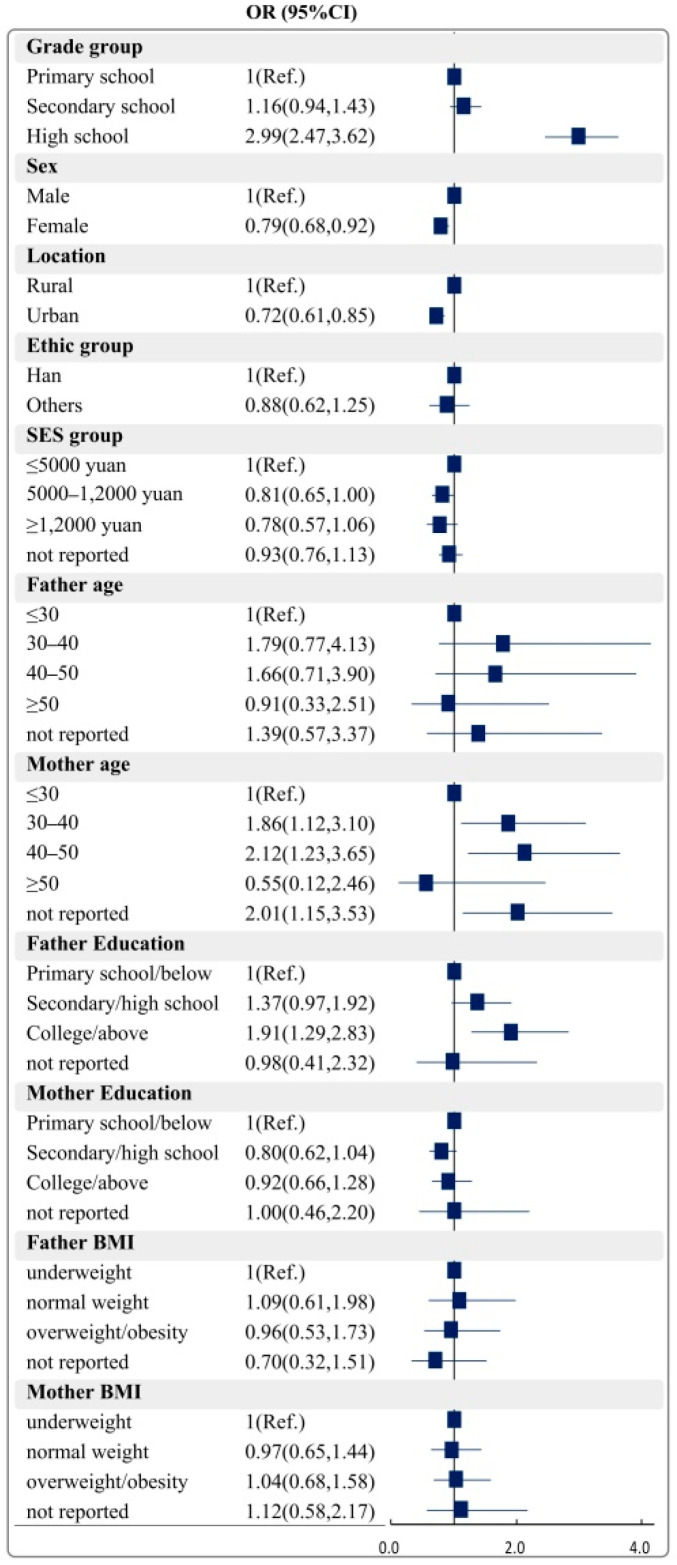
Correlates of meeting the 24 h movement guidelines among children and adolescents. BMI: body mass index; SES: social economic status; OR: odds ratio; 95%CI: 95%confidence interval.

**Figure 4 nutrients-14-02822-f004:**
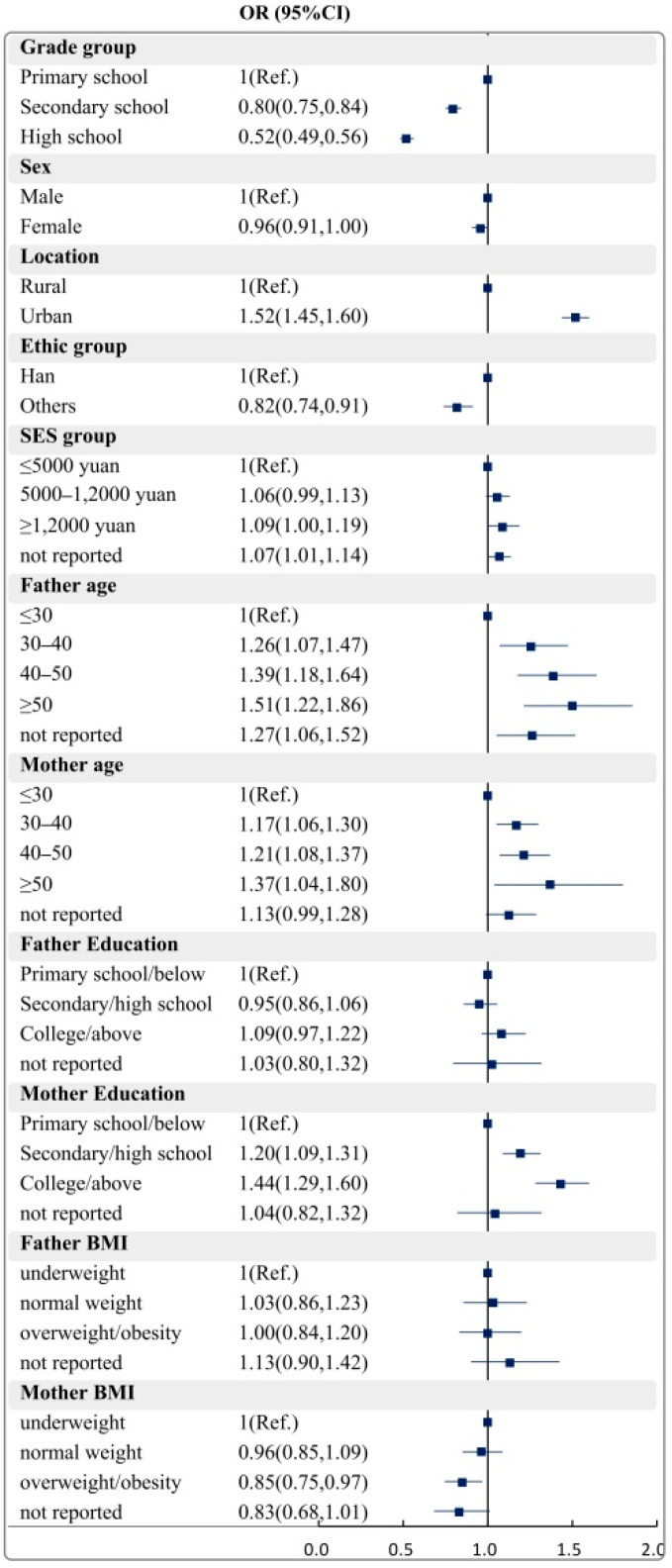
Correlates of meeting the dietary guidelines among children and adolescents. BMI: body mass index; SES: social economic status; OR: odds ratio; 95%CI: 95%confidence interval.

**Table 1 nutrients-14-02822-t001:** General characteristics of the study population.

Variables	Overall (*n* = 34,887)			Boys (*n* = 18,074)			Girls (*n* = 16,813)			*p*
	N	Freq/Mean	95%CI/SD	N	Freq/Mean	95%CI/SD	N	Freq/Mean	95%CI/SD	
**Regions**										
Central China	3633	10.40%	(10.1%, 10.7%)	1937	10.70%	(10.3%, 11.2%)	1696	10.10%	(9.6%, 10.5%)	0.007
Northwest China	1036	3.00%	(2.8%, 3.2%)	519	2.90%	(2.6%, 3.1%)	517	3.10%	(2.8%, 3.3%)	
North China	6149	17.60%	(17.2%, 18%)	3222	17.80%	(17.3%, 18.4%)	2927	17.40%	(16.8%, 18%)	
Southwest China	6288	18.00%	(17.6%, 18.4%)	3192	17.70%	(17.1%, 18.2%)	3096	18.40%	(17.8%, 19%)	
Northeast China	5954	17.10%	(16.7%, 17.5%)	3016	16.70%	(16.1%, 17.2%)	2938	17.50%	(16.9%, 18.1%)	
East China	6951	19.90%	(19.5%, 20.3%)	3697	20.50%	(19.9%, 21%)	3254	19.40%	(18.8%, 20%)	
South China	4876	14.00%	(13.6%, 14.3%)	2491	13.80%	(13.3%, 14.3%)	2385	14.20%	(13.7%, 14.7%)	
**Age**	34,887	11.42	3.22	18,074	11.32	3.2	16,813	11.52	3.23	<0.001
**BMI (kg/m^2^)**	34,887	18.76	3.84	18,074	19.09	4.03	16,813	18.41	3.59	<0.001
**Grade**										<0.001
Primary school	18,664	53.50%	(53%, 54%)	10,009	55.40%	(54.7%, 56.1%)	8655	51.50%	(50.7%, 52.2%)	
Secondary school	8944	25.60%	(25.2%, 26.1%)	4560	25.20%	(24.6%, 25.9%)	4384	26.10%	(25.4%, 26.7%)	
High school	7279	20.90%	(20.4%, 21.3%)	3505	19.40%	(18.8%, 20%)	3774	22.40%	(21.8%, 23.1%)	
**Ethnicity**										0.128
Non-Han	1837	5.30%	(5%, 5.5%)	920	5.10%	(4.8%, 5.4%)	917	5.50%	(5.1%, 5.8%)	
Han	33,050	94.70%	(94.5%, 95%)	17,154	94.90%	(94.6%, 95.2%)	15,896	94.50%	(94.2%, 94.9%)	
**Residence**										
Urban	15,705	45.00%	(44.5%, 45.5%)	8086	44.70%	(44%, 45.5%)	7619	45.30%	(44.6%, 46.1%)	0.278
Rural	19,182	55.00%	(54.5%, 55.5%)	9988	55.30%	(54.5%, 56%)	9194	54.70%	(53.9%, 55.4%)	
**Family structure**										<0.001
Single child	23,969	68.70%	(68.2%, 69.2%)	13,190	73.00%	(72.3%, 73.6%)	10,779	64.10%	(63.4%, 64.8%)	
Non-single child	10,918	31.30%	(30.8%, 31.8%)	4884	27.00%	(26.4%, 27.7%)	6034	35.90%	(35.2%, 36.6%)	
**Father’s age**										<0.001
≤30	890	2.60%	(2.4%, 2.7%)	489	2.70%	(2.5%, 2.9%)	401	2.40%	(2.2%, 2.6%)	
30–40	17,526	50.20%	(49.7%, 50.8%)	9064	50.10%	(49.4%, 50.9%)	8462	50.30%	(49.6%, 51.1%)	
40–50	10,976	31.50%	(31%, 32%)	5484	30.30%	(29.7%, 31%)	5492	32.70%	(32%, 33.4%)	
>50	1017	2.90%	(2.7%, 3.1%)	446	2.50%	(2.2%, 2.7%)	571	3.40%	(3.1%, 3.7%)	
not reported	4478	12.80%	(12.5%, 13.2%)	2591	14.30%	(13.8%, 14.9%)	1887	11.20%	(10.8%, 11.7%)	
**Mother’s age**										<0.001
≤30	2184	6.30%	(6%, 6.5%)	1111	6.10%	(5.8%, 6.5%)	1073	6.40%	(6%, 6.8%)	
30–40	18,784	53.80%	(53.3%, 54.4%)	9558	52.90%	(52.2%, 53.6%)	9226	54.90%	(54.1%, 55.6%)	
40–50	5899	16.90%	(16.5%, 17.3%)	2825	15.60%	(15.1%, 16.2%)	3074	18.30%	(17.7%, 18.9%)	
>50	303	0.90%	(0.8%, 1%)	120	0.70%	(0.6%, 0.8%)	183	1.10%	(0.9%, 1.3%)	
not reported	7717	22.10%	(21.7%, 22.6%)	4460	24.70%	(24.1%, 25.3%)	3257	19.40%	(18.8%, 20%)	
**Father’s education**										<0.001
Primary and below	2219	6.40%	(6.1%, 6.6%)	1119	6.20%	(5.8%, 6.5%)	1100	6.50%	(6.2%, 6.9%)	
Secondary/high school	20,630	59.10%	(58.6%, 59.6%)	10,651	58.90%	(58.2%, 59.6%)	9979	59.40%	(58.6%, 60.1%)	
College and above	9402	26.90%	(26.5%, 27.4%)	4763	26.40%	(25.7%, 27%)	4639	27.60%	(26.9%, 28.3%)	
not reported	2636	7.60%	(7.3%, 7.8%)	1541	8.50%	(8.1%, 8.9%)	1095	6.50%	(6.1%, 6.9%)	
**Mother’s education**										<0.001
Primary and below	3113	8.90%	(8.6%, 9.2%)	1596	8.80%	(8.4%, 9.3%)	1517	9.00%	(8.6%, 9.5%)	
Secondary/high school	20,536	58.90%	(58.3%, 59.4%)	10,536	58.30%	(57.6%, 59%)	10,000	59.50%	(58.7%, 60.2%)	
College and above	8552	24.50%	(24.1%, 25%)	4353	24.10%	(23.5%, 24.7%)	4199	25.00%	(24.3%, 25.6%)	
not reported	2686	7.70%	(7.4%, 8%)	1589	8.80%	(8.4%, 9.2%)	1097	6.50%	(6.2%, 6.9%)	
**Household income/month**										<0.001
<CNY 5000	9431	27.00%	(26.6%, 27.5%)	4655	25.80%	(25.1%, 26.4%)	4776	28.40%	(27.7%, 29.1%)	
CNY 5000–12,000	8779	25.20%	(24.7%, 25.6%)	4549	25.20%	(24.5%, 25.8%)	4230	25.20%	(24.5%, 25.8%)	
≥CNY 12,000	3209	9.20%	(8.9%, 9.5%)	1654	9.20%	(8.7%, 9.6%)	1555	9.20%	(8.8%, 9.7%)	
not reported	13,468	38.60%	(38.1%, 39.1%)	7216	39.90%	(39.2%, 40.6%)	6252	37.20%	(36.5%, 37.9%)	
**Family Chronic disease history**										<0.001
no	23,064	66.10%	(65.6%, 66.6%)	11,688	64.70%	(64%, 65.4%)	11,376	67.70%	(67%, 68.4%)	
yes	3333	9.60%	(9.2%, 9.9%)	1654	9.20%	(8.7%, 9.6%)	1679	10.00%	(9.5%, 10.4%)	
not reported	8490	24.30%	(23.9%, 24.8%)	4732	26.20%	(25.5%, 26.8%)	3758	22.40%	(21.7%, 23%)	
**Child’s Weight status**										<0.001
underweight	3956	11.30%	(11%, 11.7%)	1693	9.40%	(8.9%, 9.8%)	2263	13.50%	(13%, 14%)	
normal weight	23,222	66.60%	(66.1%, 67.1%)	11,478	63.50%	(62.8%, 64.2%)	11,744	69.90%	(69.2%, 70.5%)	
overweight/obesity	7709	22.10%	(21.7%, 22.5%)	4903	27.10%	(26.5%, 27.8%)	2806	16.70%	(16.1%, 17.3%)	
**Father’s Weight status**										<0.001
underweight	581	1.70%	(1.5%, 1.8%)	278	1.50%	(1.4%, 1.7%)	303	1.80%	(1.6%, 2%)	
normal weight	14,006	40.10%	(39.6%, 40.7%)	7015	38.80%	(38.1%, 39.5%)	6991	41.60%	(40.8%, 42.3%)	
overweight/obesity	14,352	41.10%	(40.6%, 41.7%)	7361	40.70%	(40%, 41.4%)	6991	41.60%	(40.8%, 42.3%)	
not reported	5948	17.00%	(16.7%, 17.4%)	3420	18.90%	(18.4%, 19.5%)	2528	15.00%	(14.5%, 15.6%)	
**Mother’s Weight status**										<0.001
underweight	1280	3.70%	(3.5%, 3.9%)	702	3.90%	(3.6%, 4.2%)	578	3.40%	(3.2%, 3.7%)	
normal weight	20,643	59.20%	(58.7%, 59.7%)	10,396	57.50%	(56.8%, 58.2%)	10,247	60.90%	(60.2%, 61.7%)	
overweight/obesity	7022	20.10%	(19.7%, 20.6%)	3519	19.50%	(18.9%, 20.1%)	3503	20.80%	(20.2%, 21.5%)	
not reported	5942	17.00%	(16.6%, 17.4%)	3457	19.10%	(18.6%, 19.7%)	2485	14.80%	(14.2%, 15.3%)	

Abbreviation: BMI: body mass index. SD: standard deviation; 95%CI: 95% confidence interval.

**Table 2 nutrients-14-02822-t002:** Prevalence of meeting the 24-hour movement guidelines.

24 h Movement Guidelines	Overall (*n* = 34,887)			
	Meeting *N* (%)		95%CI	
MVPA	12,466(35.7%)		(35.2%, 36.2%)	
ST	6215(17.8%)		(17.4%, 18.2%)	
SLP	12,704(36.4%)		(35.9%, 36.9%)	
MVPA + ST	1945(5.6%)		(5.3%, 5.8%)	
MVPA + SLP	4646(13.3%)		(13%, 13.7%)	
ST+SLP	2183(6.3%)		(6%, 6.5%)	
All	716(2.1%)		(1.9%, 2.2%)	
	Children 6–11 years old (*n* = 18,979)		Adolescent 12–17-year-olds (*n* = 15,908)	
	Meeting *N* (%)	95%CI	Meeting *N* (%)	95%CI
MVPA	6806(35.9%)	(35.2%, 36.5%)	5660(35.6%)	(34.8%, 36.3%)
ST	2255(11.9%)	(11.4%, 12.3%)	3960(24.9%)	(24.2%, 25.6%) ***
SLP	6875(36.2%)	(35.5%, 36.9%)	5829(36.6%)	(35.9%, 37.4%)
MVPA + ST	680(3.6%)	(3.3%, 3.9%)	1265(8%)	(7.5%, 8.4%) ***
MVPA + SLP	2573(13.6%)	(13.1%, 14%)	2073(13%)	(12.5%, 13.6%) ***
ST+SLP	830(4.4%)	(4.1%, 4.7%)	1353(8.5%)	(8.1%, 8.9%)
All	281(1.5%)	(1.3%, 1.7%)	435(2.7%)	(2.5%, 3%)
	Boys (*n* = 18,074)		Girls (*n* = 16,813)	
	Meeting *N* (%)	95%CI	Meeting *N* (%)	95%CI
MVPA	7222(40%)	(39.2%, 40.7%)	5244(31.2%)	(30.5%, 31.9%) ***
ST	2869(15.9%)	(15.3%, 16.4%)	3346(19.9%)	(19.3%, 20.5%) ***
SLP	6755(37.4%)	(36.7%, 38.1%)	5949(35.4%)	(34.7%, 36.1%) ***
MVPA + ST	1060(5.9%)	(5.5%, 6.2%)	885(5.3%)	(4.9%, 5.6%) *
MVPA + SLP	2760(15.3%)	(14.8%, 15.8%)	1886(11.2%)	(10.7%, 11.7%) ***
ST+SLP	1023(5.7%)	(5.3%, 6%)	1160(6.9%)	(6.5%, 7.3%) ***
All	400(2.2%)	(2%, 2.4%)	316(1.9%)	(1.7%, 2.1%) *

* *p* < 0.05, *** *p* < 0.001. Abbreviation: 95%CI = 95% confidence interval; MVPA = moderate-to-vigorous physical activity; OR = odds ratio; SLP = sleep duration; ST = screen time.

**Table 3 nutrients-14-02822-t003:** Prevalence of meeting the various dietary guidelines.

Dietary Guidelines	Overall (*n* = 34,887)			
	Meeting *N* (%)		95%CI	
Breakfast	29,439(84.4%)		(84%, 84.8%)	
Fruit	9109(26.1%)		(25.7%, 26.6%)	
Vegetable	2856(8.2%)		(7.9%, 8.5%)	
Meat intake	7344(21.1%)		(20.6%, 21.5%)	
Soft drink intake	11,225(32.2%)		(31.7%, 32.7%)	
Milk intake	15,370(44.1%)		(43.5%, 44.6%)	
	Children 6–11 years old (*n* = 18,979)		Adolescent 12–17-year-olds (*n* = 15,908)	
	Meeting *N* (%)	95%CI	Meeting *N* (%)	95%CI
Breakfast	17,266(91%)	(90.6%, 91.4%)	12,173(76.5%)	(75.9%, 77.2%) ***
Fruit	5328(28.1%)	(27.4%, 28.7%)	3781(23.8%)	(23.1%, 24.4%) ***
Vegetable	1449(7.6%)	(7.3%, 8%)	1407(8.8%)	(8.4%, 9.3%) ***
Meat intake	3556(18.7%)	(18.2%, 19.3%)	3788(23.8%)	(23.2%, 24.5%) ***
Soft drink intake	6909(36.4%)	(35.7%, 37.1%)	4316(27.1%)	(26.4%, 27.8%) ***
Milk intake	9121(48.1%)	(47.3%, 48.8%)	6249(39.3%)	(38.5%, 40%) ***
	Boys (*n* = 18,074)		Girls (*n* = 16,813)	
	Meeting *N* (%)	95%CI	Meeting *N* (%)	95%CI
Breakfast	15,291(84.6%)	(84.1%, 85.1%)	14,148(84.1%)	(83.6%, 84.7%)
Fruit	4539(25.1%)	(24.5%, 25.7%)	4570(27.2%)	(26.5%, 27.9%) ***
Vegetable	1550(8.6%)	(8.2%, 9%)	1306(7.8%)	(7.4%, 8.2%) **
Meat intake	4608(25.5%)	(24.9%, 26.1%)	2736(16.3%)	(15.7%, 16.8%) ***
Soft drink intake	5206(28.8%)	(28.1%, 29.5%)	6019(35.8%)	(35.1%, 36.5%) ***
Milk intake	8202(45.4%)	(44.7%, 46.1%)	7168(42.6%)	(41.9%, 43.4%) ***

** *p* < 0.01, *** *p* < 0.001. Abbreviation: 95%CI = 95% confidence interval; MVPA = moderate-to-vigorous physical activity; OR: odds ratio; SLP = sleep duration; ST = screen time.

**Table 4 nutrients-14-02822-t004:** Logistic regression analysis of the associations between meeting 24 h movement and dietary guidelines and weight status.

	Model 1				Model 2			
	Underweight		Overweight/Obesity		Underweight		Overweight/Obesity	
	OR (95%CI)	*p*	OR (95%CI)	*p*	95%CI	*p*	95%CI	*p*
None	1.22(0.95,1.58)	0.123	1.26(1.03,1.54)	0.026	1.23(0.95,1.58)	0.117	1.22(1.00,1.49)	0.053
MVPA	1.09(0.84,1.41)	0.538	1.33(1.09,1.64)	0.006	1.10(0.85,1.43)	0.479	1.30(1.06,1.60)	0.011
ST	1.09(0.83,1.44)	0.537	1.35(1.09,1.67)	0.006	1.08(0.82,1.42)	0.589	1.32(1.06,1.64)	0.012
SLP	1.35(1.04,1.75)	0.023	1.18(0.96,1.45)	0.114	1.35(1.04,1.75)	0.023	1.15(0.94,1.41)	0.186
ST+MVPA	1.20(0.88,1.63)	0.248	1.27(1.00,1.61)	0.052	1.20(0.88,1.62)	0.253	1.25(0.98,1.59)	0.068
SLP+MVPA	1.04(0.79,1.36)	0.780	1.24(1.01,1.53)	0.043	1.05(0.80,1.37)	0.730	1.22(0.99,1.5)	0.067
ST+SLP	1.29(0.96,1.74)	0.087	1.02(0.80,1.30)	0.865	1.28(0.95,1.71)	0.106	1.00(0.79,1.27)	0.999
MVPA+SLP+ST	1		1		1		1	
Breakfast	0.95(0.86,1.05)	0.280	1.07(0.99,1.16)	0.070	0.94(0.85,1.04)	0.248	1.07(0.99,1.15)	0.097
Fruit	1.03(0.96,1.12)	0.412	0.94(0.91,1.07)	0.051	1.02(0.94,1.11)	0.637	0.93(0.87,0.99)	0.022
Vegetable	0.99(0.87,1.12)	0.855	0.95(0.87,1.05)	0.319	0.97(0.86,1.10)	0.659	0.96(0.87,1.05)	0.478
Meat intake	0.98(0.90,1.07)	0.712	1.01(0.94,1.08)	0.817	0.98(0.90,1.07)	0.596	1.01(0.95,1.08)	0.727
Soft drink intake	0.91(0.85,0.98)	0.010	1.01(0.96,1.07)	0.665	0.91(0.84,0.98)	0.010	1.01(0.95,1.07)	0.850
Milk intake	1.08(1.01,1.64)	0.024	1.15(1.09,1.22)	<0.001	1.07(1.00,1.15)	0.063	1.16(1.10,1.23)	<0.001

Notes: Model 1 was adjusted for age, gender, grade, parental education level, parents’ age, family income, family chronic disease history, residence, location, family structure and parents’ weight status. For movement variables, Model 2 was further adjusted for dietary variables, including breakfast, fruit, vegetable, meat, soft drink and milk intake. For dietary variables, Model 2 was further adjusted for movement variables, including MVPA, SLP and ST. In addition, those meeting the dietary guidelines were set as the reference group. Abbreviation: 95%CI = 95% confidence interval; MVPA = moderate-to-vigorous physical activity; OR = odds ratio; SLP = sleep duration; ST = screen time.

## Data Availability

The data are not publicly available due to privacy restrictions.

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
