# Peer review of "Meeting 24-Hour Movement and Dietary Guidelines: Prevalence, Correlates and Association with Weight Status among Children and Adolescents: A National Cross-Sectional Study in China"

_nutrients, 2022, doi:10.3390/nu14142822_

Round 1

Reviewer 1 Report

Dear Authors,

The issues addressed in the article are very interesting and of relevance to this reviewer. In many parts of the world, as exemplified by China, we are dealing with a "double burden" of underweight and overweight/obesity. Overweight and obesity is an increasingly common condition observed in children and adolescents. It results from an excess of calories supplied with food in relation to the current demand of the body. The vast majority of obesity and underweight cases in children are caused by an inappropriate lifestyle and poor diet. The basis of obesity treatment is to modify the daily menu and increase physical activity.  Hunger is one of the main causes of malnutrition and nutrient deficiencies in poor, economically underdeveloped countries, and affects large social groups or entire populations there. However, people also go hungry in developed and developing countries. Malnutrition is most often caused by a lack of food resources due to: lack of work, failure to coping with life problems and social alienation, strong addiction to addictions.

Meeting the 24-h movement and dietary guidelines was associated with weight status in Chinese children and adolescents. The study findings provide empirical evidence for future health interventions and policy-making in the fight against childhood obesity.

In my opinion:

- The manuscript uses a clear introduction,

- The material and research methods were described in detail;

- The study was properly planned;

- Appropriate analytical methods were selected to analyse the results;

- The results of the analyses were presented in an understandable verbal and tabular form.

Please respond to my concerns:

1.       Do the authors consider the Canadian 24-h Movement Guidelines to be a good research tool for studying the problem of obesity and underweight for Chinese children?

2.       What measures is China taking to address regional differences in the prevalence of obesity and underweight in children and adults?

3.       Please list 3 main causes of overweight/obesity and underweight in Chinese children each.

With sincerely,

Author Response

Point 1: Do the authors consider the Canadian 24-h Movement Guidelines to be a good research tool for studying the problem of obesity and underweight for Chinese children?

Response 1: Thank you for your comments. Yes, we think that the Canadian 24-h guidelines is a good research tool for comprehensively studying the problem of obesity and underweight for Chinese children, and we have elaborated more on the Canadian 24-h Movement Guidelines.

“The Canadian 24-Hour Movement Guidelines integrated recommendations that respect the intuitive and integrative nature of children's physical activity. The Guidelines recommend that school-aged children (5-17 years) should perform at least 60 minutes of moderate-to-vigorous physical activity (MVPA) per day, sleep 9 to 11 hours per day, and keep screen behavior to be less than two hours per day. The development process of the guidelines was guided by the Guidelines for Research Evaluation (AGREE) II instrument and systematic reviews of evidence informing the guidelines were assessed according the Grading of Recommendations Assessment, Development, and Evaluation (GRADE) approach. Four systematic reviews (physical activity, sedentary behaviour, sleep, and integrated behaviours) and several health indicators were recognized by expert consensus. The Canadian 24-Hour Movement Guidelines integrated recommendations have been applied as the research paradigm and have been used by many researchers in their research. (please see Lines 67-79)

Point 2: What measures is China taking to address regional differences in the prevalence of obesity and underweight in children and adults?

Response 2: Thank you for your valuable comment. We have made the following explanation according to your concerns.

“The Chinese government has formulated a rural vitalization policy and anti-poverty relocation and settlement program (ARSP) to narrow the economic development gap between urban and rural areas and solve the poverty trap in western regions[1,2], helping reduce the risk of underweight among children and teenagers in poor areas. In addition, prevalence of obesity are rising among children and adolescents in developed region[3]. Therefore, panel of chinese experts developed a consensus statement, which stressed the importance of promoting physical activity and provided suggestions for promoting physical activity among school-age children and adolescents to prevent childhood and adolescent diseases including obesity and achieve Healthy China 2030 goals[4]. The Healthy China 2030 national strategy is the most remarkable measures to control the obesity and underweight problems, which was initiated in 2016[5]. During the past decades, China have made many efforts in tackling the obesity and underweight problems[6]. Moreover, the Dietary Guidelines for Chinese (DGC-2016) was released in 2016, with the aim of developing dietary guidelines suitable for Chinese people and help them develop healthier and more scientific eating habits [7,8].“ (please see Lines 313-327)

Point 3: Please list 3 main causes of overweight/obesity and underweight in Chinese children each.With sincerely

Response 3: Thank you for your comments. For the main causes of overweight/obesity and underweight in Chinese Children, we have reviewed more papers and listed 3 main causes of overweight/obesity and underweight as following.

Weight status of Chinese children is closely associated with dietary behaviors, physical activity,and sedentary behavior. Regular physical activity have a protective effect on healthy weight status among children and adolescents, while sedentary behavior (ST) generally has a negative impact[9,10]. Previous studies indicated that altering physical activity, and sedentary behavior were effective in improving overweight and obesity in children and adolescents[11-13]. Additionally, a previous study also found a significant lower level of physical activity in underweight boys compared with normal weight[14]. Dietary behaviors have also been demonstrated to be important determinants of weight status [15]. Eating too many high-calorie food(such as fast food and soft drink) is associated with an increased risk of obesity [16]. Children who have an unbalanced diet or picky eating behaviour are more likely to be underweight or malnourished[17]. (please see Lines 60-64,Lines 90-92)

  1. Li, C.; Li, M. The Policy Information Gap and Resettlers' Well-Being: Evidence from the Anti-Poverty Relocation and Resettlement Program in China. International journal of environmental research and public health 2020, 17, doi:10.3390/ijerph17082957.
  2. Hou, D.; Wang, X. Measurement of Agricultural Green Development Level in the Three Provinces of Northeast China Under the Background of Rural Vitalization Strategy. Frontiers in public health 2022, 10, 824202, doi:10.3389/fpubh.2022.824202.
  3. Gan, X.; Xu, W.; Yu, K. Economic Growth and Weight of Children and Adolescents in Urban Areas: A Panel Data Analysis on Twenty-Seven Provinces in China, 1985-2014. Childhood obesity (Print) 2020, 16, 86-93, doi:10.1089/chi.2019.0151.
  4. Chen, P.; Wang, D.; Shen, H.; Yu, L.; Gao, Q.; Mao, L.; Jiang, F.; Luo, Y.; Xie, M.; Zhang, Y., et al. Physical activity and health in Chinese children and adolescents: expert consensus statement (2020). British journal of sports medicine 2020, 54, 1321-1331, doi:10.1136/bjsports-2020-102261.
  5. Wang, Y.; Zhao, L.; Gao, L.; Pan, A.; Xue, H. Health policy and public health implications of obesity in China. The lancet. Diabetes & endocrinology 2021, 9, 446-461, doi:10.1016/s2213-8587(21)00118-2.
  6. Wang, H.; Zhai, F. Programme and policy options for preventing obesity in China. Obesity reviews : an official journal of the International Association for the Study of Obesity 2013, 14 Suppl 2, 134-140, doi:10.1111/obr.12106.
  7. Yuan, Y.Q.; Li, F.; Dong, R.H.; Chen, J.S.; He, G.S.; Li, S.G.; Chen, B. The Development of a Chinese Healthy Eating Index and Its Application in the General Population. Nutrients 2017, 9, doi:10.3390/nu9090977.
  8. Kadam, I.; Neupane, S.; Wei, J.; Fullington, L.A.; Li, T.; An, R.; Zhao, L.; Ellithorpe, A.; Jiang, X.; Wang, L. A Systematic Review of Diet Quality Index and Obesity among Chinese Adults. Nutrients 2021, 13, doi:10.3390/nu13103555.
  9. Carson, V.; Chaput, J.P.; Janssen, I.; Tremblay, M.S. Health associations with meeting new 24-hour movement guidelines for Canadian children and youth. Preventive medicine 2017, 95, 7-13, doi:10.1016/j.ypmed.2016.12.005.
  10. Saunders, T.J.; Gray, C.E.; Poitras, V.J.; Chaput, J.P.; Janssen, I.; Katzmarzyk, P.T.; Olds, T.; Connor Gorber, S.; Kho, M.E.; Sampson, M., et al. Combinations of physical activity, sedentary behaviour and sleep: relationships with health indicators in school-aged children and youth. Applied physiology, nutrition, and metabolism = Physiologie appliquee, nutrition et metabolisme 2016, 41, S283-293, doi:10.1139/apnm-2015-0626.
  11. Ip, P.; Ho, F.K.; Louie, L.H.; Chung, T.W.; Cheung, Y.F.; Lee, S.L.; Hui, S.S.; Ho, W.K.; Ho, D.S.; Wong, W.H., et al. Childhood Obesity and Physical Activity-Friendly School Environments. The Journal of pediatrics 2017, 191, 110-116, doi:10.1016/j.jpeds.2017.08.017.
  12. Al-Khudairy, L.; Loveman, E.; Colquitt, J.L.; Mead, E.; Johnson, R.E.; Fraser, H.; Olajide, J.; Murphy, M.; Velho, R.M.; O'Malley, C., et al. Diet, physical activity and behavioural interventions for the treatment of overweight or obese adolescents aged 12 to 17 years. The Cochrane database of systematic reviews 2017, 6, Cd012691, doi:10.1002/14651858.Cd012691.
  13. Mead, E.; Brown, T.; Rees, K.; Azevedo, L.B.; Whittaker, V.; Jones, D.; Olajide, J.; Mainardi, G.M.; Corpeleijn, E.; O'Malley, C., et al. Diet, physical activity and behavioural interventions for the treatment of overweight or obese children from the age of 6 to 11 years. The Cochrane database of systematic reviews 2017, 6, Cd012651, doi:10.1002/14651858.Cd012651.
  14. Kantanista, A.; Osiński, W. Underweight in 14 to 16 year-old girls and boys: prevalence and associations with physical activity and sedentary activities. Annals of agricultural and environmental medicine : AAEM 2014, 21, 114-119.
  15. Keast, D.R.; Hill Gallant, K.M.; Albertson, A.M.; Gugger, C.K.; Holschuh, N.M. Associations between yogurt, dairy, calcium, and vitamin D intake and obesity among U.S. children aged 8-18 years: NHANES, 2005-2008. Nutrients 2015, 7, 1577-1593, doi:10.3390/nu7031577.
  16. Roblin, L. Childhood obesity: food, nutrient, and eating-habit trends and influences. Applied physiology, nutrition, and metabolism = Physiologie appliquee, nutrition et metabolisme 2007, 32, 635-645, doi:10.1139/h07-046.
  17. Lee, G.; Ham, O.K. Factors Affecting Underweight and Obesity Among Elementary School Children in South Korea. Asian nursing research 2015, 9, 298-304, doi:10.1016/j.anr.2015.07.004.

Reviewer 2 Report

The reviewed manuscript aimed to evaluate the prevalence of complying with 24-h movement and dietary guidelines among Chinese children and adolescents and its association with weight status.

Some of the associations found are striking, such as the association between not meeting the recommendations for fruit consumption and the risk of being overweight/obese. The authors discuss this aspect and suggest that the sugar content of the fruit could lead to obesity. Could other factors in the studied sample contribute to this finding? Some sociodemographic characteristics? Given how the information was obtained, the authors’ suggestion that different ways of consuming fruits could influence this association is tentative and presented without supporting citations (line 342). It is suggested that the authors can elaborate further in this regard.

In the current form of Tables 2 and 3, they do not allow an easy comparison of data between age groups. Including age groups as columns, not in rows, might facilitate a better understanding of readers.

Minors:

- Several citations are included in a disorderly manner, in different formats, throughout the manuscript (lines 43, 88, etc.). Please review all citations.

- Line 47: The abbreviation must appear after its definition

- Line 137: include more detailed information about the instrument used to measure physical activity

- In Results: when indicating “significant” please include the level of statistical significance reached

Authors must check the English grammar throughout the manuscript. A review by a native English speaker is required.

Author Response

Point 1: Some of the associations found are striking, such as the association between not meeting the recommendations for fruit consumption and the risk of being overweight/obese. The authors discuss this aspect and suggest that the sugar content of the fruit could lead to obesity. Could other factors in the studied sample contribute to this finding? Some sociodemographic characteristics? Given how the information was obtained, the authors’ suggestion that different ways of consuming fruits could influence this association is tentative and presented without supporting citations (line 342). It is suggested that the authors can elaborate further in this regard.

Response 1: Thank you for your comments. We have made the following explanation according to your suggestions.

“Most of previous studies did not distinguish the ways to consuming fruits, such as fresh whole fruits, 100% fruit Juice, blended juice or tinned fruit, leading to inconsistent results on association between fruit consumption and weight. Fruit contains various kinds of simple sugars (glucose, fructose, etc.), which are well known to lead to obesity. In addition, Studies have confirmed that tinned fruit and blended juice contain added sugars, which may increase the risk of obesity. This support our results. Additionally, previous studies find that low socioeconomic status and fruit intakes would be associated with adolescent overweight/obesity. Fruit intakes would mediate the relationship between low socioeconomic status and adolescent overweight/obesity. These findings deserve further exploration.” (please see Lines 376-385)

Point 2: In the current form of Tables 2 and 3, they do not allow an easy comparison of data between age groups. Including age groups as columns, not in rows, might facilitate a better understanding of readers.

Response 2: Thank you for your suggestions. We have modified the form of Tables 2 and 3 according to your suggestions.

Point 3: Several citations are included in a disorderly manner, in different formats, throughout the manuscript (lines 43, 88, etc.). Please review all citations.

Response 3: Thank you for your comments. We have reviewed all citations and modified the wrong citations.

Point 4: The abbreviation must appear after its definition.

Response 4: Thank you for your suggestions. We have changed the order between the abbreviation and its definition according to your suggestions.

“Previous studies have also shown that a higher body mass index (BMI) during adolescence was associated with a higher risk of malignancies in adulthood”. (please see Lines 46)

Point 5: include more detailed information about the instrument used to measure physical activity.

Response 5: Thank you for your comments. We have added more details of the instrument used to measure physical activity according to your suggestions (Page 3-4).

“Frequency (days each week) and duration (specific hours spent on these activities for each of these days) of vigorous PA (obviously increase ones breathing and heart rate, such as running, basketball and football et al) and moderate PA(to some extent increase one’s breathing and heart rate, such as table tennis, biking, badminton et al) were investigated[1]. (please see Lines 155-159)

Point 6: when indicating “significant” please include the level of statistical significance reached.

Response 6: Thank you for your comments. We have checked the full text and included the level of statistical significance reached when indicating “significant”.

Point 7: Authors must check the English grammar throughout the manuscript. A review by a native English speaker is required.

Response 7: Thank you for your suggestions. We have checked the English grammar thoroughly by Prof. Julien S. Baker, who is a native English speaker.

  1. Yang, Y.D.; Xie, M.; Zeng, Y.; Yuan, S.; Tang, H.; Dong, Y.; Zou, Z.; Dong, B.; Wang, Z.; Ye, X., et al. Impact of short-term change of adiposity on risk of high blood pressure in children: Results from a follow-up study in China. PloS One 2021, 16, e0257144, doi:10.1371/journal.pone.0257144.
